# Photocatalytic Degradation of Malachite Green Dye from Aqueous Solution Using Environmentally Compatible Ag/ZnO Polymeric Nanofibers

**DOI:** 10.3390/polym13132033

**Published:** 2021-06-22

**Authors:** Marwa F. Elkady, Hassan Shokry Hassan

**Affiliations:** 1Fabrication Technology Department, Advanced Technology and New Materials Researches Institute, City of Scientific Researches and Technological Applications (SRTA-City), New Borg El-Arab City, Alexandria 21934, Egypt; 2Chemical and Petrochemical Engineering Department, Egypt-Japan University of Science and Technology, New Borg El-Arab City, Alexandria 21934, Egypt; 3Electronic Materials Researches Department, Advanced Technology and New Materials Researches Institute, City of Scientific Researches and Technological Applications (SRTA-City), New Borg El-Arab City, Alexandria 21934, Egypt; 4Environmental Engineering Department, Egypt-Japan University of Science and Technology, New Borg El-Arab City, Alexandria 21934, Egypt

**Keywords:** photocatalytic, malachite green, wastewater remediation, ZnO/CA/Ppy, composite nanofiber, silver immobilization

## Abstract

An efficient, environmentally compatible and highly porous, silver surface-modified photocatalytic zinc oxide/cellulose acetate/ polypyrrole ZnO/CA/Ppy hybrid nanofibers matrix was fabricated using an electrospinning technique. Electrospinning parameters such as solution flow rate, applied voltage and the distance between needles to collector were optimized. The optimum homogenous and uniform ZnO/CA/Ppy polymeric composite nanofiber was fabricated through the dispersion of 0.05% wt ZnO into the dissolved hybrid polymeric solution with an average nanofiber diameter ranged between 125 and 170 nm. The fabricated ZnO-polymeric nanofiber was further surface-immobilized with silver nanoparticles to enhance its photocatalytic activity through the reduction of the nanofiber bandgap. A comparative study between ZnO polymeric nanofiber before and after silver immobilization was investigated using X-ray diffraction (XRD), scanning electron microscope (SEM), transmission electron microscopy (TEM), Fourier transform infrared (FTIR) and thermal gravimetric analysis (TGA). The photocatalytic degradation efficiency of the two different prepared nanofibers before and after nanosilver immobilization for malachite green (MG) dye was compared against various experimental parameters. The optimum degradation efficiency of nanosilver surface-modified ZnO-polymeric nanofibers was recorded as 93.5% for malachite green dye after 1 h compared with 63% for ZnO-polymeric nanofibers.

## 1. Introduction

The manufacturing of fabrics by textile industries has an abysmal impact on the environment due to the huge expending of synthetic dyes that improve the BOD and COD, impair photosynthesis, hinder plant growth, provide recalcitrance and promote carcinogenicity and mutagenicity [1,2]. Besides, insatiable consumption of water running to trillions of gallons per annum and large amounts of processed dye effluents containing thousands of tons of unconsumed dyes pose a great threat to ecology and the environment [3]. Despite stringent regulations imposed on the textile industries on wastewater treatment, the outcomes have been abysmally disappointing [4]. One of the major reasons is the absence of a sustainable remediation technique, method and/or procedure for the removal of toxic dyes. Sustainability as a system addresses the issues of economics, environment and social concern [5,6].

Nanotechnology and Nanomaterials have the capability to secure sustainability in the field of industrial wastewater treatment through providing inventive technologies due to their ability to control the materials’ functions and properties to be adequate with specific demand [7,8].

Various techniques based on nanotechnology were reported in the literature for wastewater remediation, which may be broadly classified into four major categories including biological, chemical, physico-chemical and physical methods [9]. Most of these techniques are suffering from the main two limitations, which are the required remediation time for process completion and the sludge disposal after finishing the remediation process.

Photocatalytic degradation is a well-accepted favorable technique with great efficiency to completely degrade any organic compounds in the contaminated wastewater to convert them into harmless final products to human beings and the environment using a proper photocatalyst in presence of light radiation by producing highly reactive hydroxyl (OH•) radicals, which can convert water pollutants into relatively unharmful end products like CO_2_, H_2_O and other inorganic ions [10,11].

Photocatalytic technology has the ability to address the first problem concerned with remediation time with tangible solutions for the remediation of dyes from industrial effluents [12]. However, this technology still suffers from the problem of sludge disposal after the process’s completion. Thus, it is ethically imperative to address the issue to provide a congenial solution for the disposal of sludge to guarantee the sustainability of photocatalytic technology [13]. One of the alternatives is to use eco-friendly compatible photocatalytic materials, which will have less impact on the environment and ecology.

A key parameter governing the performance of the photocatalytic materials is the generation of continuous electron reflux for producing highly reactive hydroxyl (OH•) radicals responsible for dyes degradation at industrial effluents. A dangerous coloring cationic dye that requires degradation from polluted wastewater is malachite green dye (MG). This dye belongs to the triphenylmethane dyes category, which is heavily used in dye effluents and waste from industries like coloring materials, printing, leather, textiles, paper, plastic, pharmaceutical industries, medical laboratories and food industries [14]. This injurious dye can cause mutagenic, carcinogenic and teratogenicity effects to humans and animals due to the chemicals that are used to produce dyes are highly toxic, hormonal disruptors and carcinogens [15].

An efficient, environmentally compatible and highly porous photocatalytic hybrid nanofibers matrix can be fabricated simply by Electrospinning (ES) technique at low cost and high production rates. This hybrid nanofiber matrix offers outstanding photocatalytic characteristics to overcome the main two limitations stated previously at any photocatalytic degradation process which are the process operation time and the sludge produced. As the electrospun nanofibers have a very considerable surface area to magnitude proportion and high porosity that guarantee fast and high photo-degradation rate at short time intervals [16,17]. In order to solve the waste sludge problem, the materials utilized for nanofiber fabrication are suggested to be completely environmentally compatible for safe sludge disposal after finishing the remediation process [18,19]. The hybrid nanofiber matrix will be fabricated through the incorporation of zinc oxide (ZnO) nanoparticle as a semiconductor filler onto the hybrid biocompatible polymeric matrix composed of cellulose acetate (CA) and polypyrrole (Ppy). As well-known, CA is renowned as a novel biomaterial and has broad industrial applications in its fibers form. Moreover, the CA fiber is described as relatively easy to manufacture, cost-economical, has good thermal stability, is a renewable natural resource of raw material, chemical-resistant and non-toxic [20,21]. Owing to the properties of Ppy as a conductive polymer in electron transfer procedure, it has been excessively utilized to excite a rapid photo-induced charge separation and comparatively slow charge recombination in photocatalytic processes including dye degradation in contaminated industrial wastewater [22]. Additionally, (ZnO) nanopowder is categorized as a semiconductor material with great importance for its versatile properties such as wide and direct fundamental energy band gap of 3.37 eV and high exciting binding energy of 60 mV. ZnO is considered a highly stable, environmentally friendly and inexpensive material. Subsequently, it has potential applications in wastewater treatment domains, especially in the field of photocatalytic applications [23]. However, most metal oxides suffering from the problem of a recombination rate of the photogenerated electrons at their hole pairs during their operations as photocatalysts. Consequently, the incorporation of metal oxides with a noble metal such as silver nanoparticles represents the best scenario to overcome the electron recombination problem to improve the electron reflux available for the photocatalytic process. Accordingly, Ag nanoparticles will be immobilized on the surface of ZnO polymeric hybrid nanofibers to improve the photocatalytic efficiency of the fabricated ZnO/CA/Ppy hybrid nanofiber and to act as electron sinks [24,25].

This work concerns the fabrication of efficient, environmentally compatible, photocatalytic, silver-immobilized ZnO/CA/Ppy hybrid polymeric nanofibers using electrospinning technique and the assessment of the degradation efficiency of malachite green dye from simulated polluted wastewater in the presence of UV irradiation. In order to solve the sludge problem after the photocatalytic process, all constituents of the nanofiber were selected to be environmentally friendly materials. Moreover, the nano-ZnO utilized as filler at the hybrid nanofiber was green, synthesized using olive plant leaves extract to decrease the environmental impact of utilizing harmful chemicals. This fabricated eco-friendly hybrid nanofiber was assessed for the safe photocatalytic degradation of polluted dye to present a complete treatment of polluted industrial wastewater.

## 2. Materials and Methods

### 2.1. Materials

Olive plant leaves were collected from New Borg El-Arab City, Alexandria, Egypt for the biosynthesis of ZnO nanoparticles. Most of the utilized other chemicals were of analytical grade with high purity including zinc nitrate hydrate (99.999%, Sigma-Aldrich, St. Louis, MO, USA), cellulose acetate (CA) (MW 30000, Sigma-Aldrich, St. Louis, MO, USA), acetic acid (99.8%, Alfa Aser, Haverhill, MA, USA), acetone (99.5%, Sigma-Aldrich, St. Louis, MO, USA), N, N-dimethyl-formamide (DMF) (99.8%, Across Organics, Fair Lawn, NJ, USA), malachite green dye (MG) (purity 99.9%, Sigma-Aldrich, St. Louis, MO, USA), silver nitrate (99%, Sigma-Aldrich, St. Louis, MO, USA), pyrrole (99.5%, Sigma-Aldrich, St. Louis, MO, USA), ferric chloride (99.9%, Across Organics, Fair Lawn, NJ, USA), sodium hydroxide and hydrochloric acid (99.8%, Sigma-Aldrich, St. Louis, MO, USA). Freshly distilled water was used for the preparations of all aqueous solutions and washing purposes.

### 2.2. Methods

#### 2.2.1. Green Synthesis of ZnO Nanoparticles as Filler

Zinc nitrate salt (Zn (NO_3_)_2_ 6H_2_O) of 0.2 M was reduced with 30 mL of olive plant leaves extract. The extract was produced after grinding the olive plant leaves that were soaked in distilled water under vigorous stirring for 1 h at 40 °C. This extract was added to zinc nitrate salt dropwise under constant stirring at 60 °C for 1 h. The produced ZnO nanopowder materials were dried at 100 °C overnight. During the drying process, complete conversion of zinc hydroxide into zinc oxide takes place until its color is converted into a sunny white color [7].

#### 2.2.2. Synthesis of Polypyrrole (Ppy) Nanoparticles

Ppy nanopowder was synthesized using the chemical oxidation polymerization technique. 120 mL of 0.2 M ferric chloride was used as an oxidant under continuous stirring for 30 min. Then, 2 mL pyrrole monomer was added dropwise under continuous stirring for 2 h; the addition was performed slowly at a low temperature around 2–7 °C using an ice bath. The produced Ppy suspension was kept for 24 h in the dark to complete the polymerization process. The formed black precipitate of Ppy was filtered and washed several times with ethanol and distilled water to remove the unreacted pyrrole and excess of ferric chloride and any impurities present. A final black precipitate of Ppy was obtained after 12 h of vacuum drying at 30–40 °C.

### 2.3. Fabrication of ZnO Polymeric Composite Nanofiber

#### Preparation of ZnO Nanoparticles as Filler into the Polymeric Composite Solution

Ppy nanopowder was dissolved in DMF/acetone mixed solvent with a ratio of 6:7 and ZnO nanoparticle was dispersed at the dissolved polypyrrole solution. DMF was mixed with acetone to delay the evaporation of acetone during electrospinning to form a uniform nanofiber. In order to prepare a homogeneous solution, this suspension was sonicated for 5 min then stirred using a magnetic stirrer at 300 rpm for 24 h. After obtaining a homogeneous suspension from ZnO nanoparticle dispersed into Ppy, add a certain amount of cellulose acetate powder to this suspension then further sonicate for 15 min and stir using a magnetic stirrer at 300 rpm for another 24 h until the formation of a homogeneous solution through the generation of a clear colloidal solution that is ready for the electrospinning process [26].

### 2.4. Electrospinning of ZnO Polymeric Composite Nanofiber

The solution contained a certain amount of dissolved CA and Ppy composite polymers in DMF: acetone 6:7 mixed solvent and 0.1 g of dispersed ZnO nanopowder as filler. This homogeneous suspension was loaded into a syringe (5 mL) with a stainless-steel needle to start the electrospinning process. The concentration of CA polymer was varied between (8 and 16 wt%) at a fixed weight from Ppy (0.3 g). The electrospinning parameters such as applied electrical voltage (8–30 kV), the distance between the tip of the syringe needle and collector (8–20 cm), feed solution flow rate (0.2–1 mL/h) and the amount of loaded ZnO nanopowder as filler into the polymeric composite mixture at room temperature (15 ± 30 °C) were optimized to fabricate uniform composite nanofiber.

### 2.5. Surface Modification of ZnO Polymeric Composite Nanofiber by Silver Immobilization

In order to reduce the bandgap of fabricated ZnO polymeric composite nanofiber to enhance its photocatalytic activity, silver was immobilized onto the fiber surface through the following surface modification process. A small piece of the fabricated ZnO polymeric composite nanofiber with dimensions of 1 × 1 cm was immersed in 100 mL of silver nitrate (0.85 g) solution mixed with the aqueous extract of olive leaves. The nanofiber was maintained socked at the solution under gentle stirring using an orbital shaker until the reduction process of silver was completed. The small piece of surface-modified Ag/ZnO polymeric composite nanofiber small was washed 5 times with distilled water to remove any residuals then dried at 40–50 °C for 8 h.

### 2.6. Characterization of ZnO Polymeric Composite Nanofibers before and after Silver Immobilization

XRD patterns of ZnO polymeric nanofibers before and after silver immobilization were obtained using Schimadzu7000 (Schimadzu, Kyoto, Japan) diffraction operating with Cu Kα radiation (λ = 0.15406 nm), generated at 30 kV and 30 mA with a scan rate of 2 min^−1^ for 2θ values between 10 and 80 degrees. Scanning electron microscope (SEM) analysis JEOL JSM 6360LA (Akishima, Tokyo, Japan) was performed to determine the shape, size and morphology of ZnO polymeric composite nanofibers. The uniformity and homogeneity of the two prepared ZnO polymeric composite nanofibers before and after silver immobilization was confirmed using TEM JEOL JEM 1230 (Akishima, Tokyo, Japan) and the percentage of silver element immobilization was confirmed using the EDX analysis combined with the equipment of the FTIR spectrum of ZnO nanofibers before and after silver immobilization was established using Shimadzu FTIR-8400 S (Schimadzu, Kyoto, Japan) at a wavelength range of 400–4000 cm^−1^. The thermal stability of ZnO nanofibers before and after silver immobilization was evaluated by TGA Shimadzu TGA–50H (Schimadzu, Kyoto, Japan).

### 2.7. Photocatalytic Degradation of Malachite Green Dye Using Various Fabricated ZnO Polymeric Composite Nanofibers

The photocatalytic degradation of malachite green (MG) dye was carried out by contacting 25 mL of 10 ppm dye solution in a 100 mL round flask with 0.05 g from either ZnO polymeric composite nanofibers before or after silver surface immobilization. The solution was exposed to Xenon lamp 500 W (Alexandria City, Egypt) for certain time intervals under continuous stirring. The dye solution sample was withdrawn for analysis using UV spectrophotometer every 10 min during the irradiation process to determine the MG dye degradation efficiency which appears in the visible region that was measured at 624 nm.

The decolorization efficiency of MG dye through the photocatalytic process was calculated by the following formula:Efficiency % = (C_o_ − C)/C_o_ × 100(1)

The concentrations of primal and decomposed dye solution are C_o_ and C, respectively. The same experimental procedure was followed to elucidate the influence of various photo-degradation parameters including contact time (0–100 min.), catalyst dosage (0.05–0.25 g), initial dye concentrations (10–100 ppm), solution pH (2–12) and temperature (25–80 °C) on the MG dye-decolorization process. Each experiment was repeated in triplicate manner and the mean results were utilized in data analysis.

Finally, the experimental data from these studied parameters that were either recorded at the equilibrium time or over the studied time intervals were theoretically modeled to determine the suitable models describing the malachite green dye photocatalytic degradation process on the two prepared ZnO polymeric composite nanofibers before and after Ag surface immobilization.

### 2.8. Reusability of Fabricated ZnO Polymeric Composite Nanofibers

In order to improve the economic feasibility of the photocatalytic process using the prepared ZnO polymeric composite nanofibers before and after silver immobilization, the spent ZnO polymeric composites nanofibers after the photocatalytic process were regenerated through three times rinsing with 0.2 mol/L HCl solution followed by deionized water then dried at 50 °C overnight for further reuse.

## 3. Results

### 3.1. Optimization of the Fabrication of ZnO Polymeric Composite Nanofibers

Nanofibers’ diameters and their homogeneity could be enhanced by changing the different parameters during the electrospinning process such as the cellulose acetate polymer solution concentration, flow rate, electrical applied voltage, distances between the syringe needle tip and the collector and the amount of loaded ZnO nanopowder as filler.

The morphological structure of nanofibers produced at the three studied cellulose acetate (A1–A3) polymer concentrations at a constant flow rate of 0.2 mL/h, an applied voltage of 20 kV and the distance between needle tip to the collector of 12 cm was investigated at Figure 1A. It is clear that no continuous fiber was formed at 8 wt% polymer concentration (A1) and only electrospray and droplets from the polymer solution were collected on the foil (collector). As the CA polymer concentration improved from 12 wt% (A2) to 16 wt% (A3), continuous and thick fibers were obtained with the appearance of some beads. These beads disappeared with increasing CA concentration due to the improvement of electrospun solution viscosity as indicated for SEM image for sample (A3) from Figure 1(A1). On the other hand, as evident from Figure 1(A1), the increment in polymer concentration from 12% to 16% tends to decrease the fiber diameter [27]. Accordingly, the CA polymer concentration of 16% was selected as the optimum concentration for further optimization of the remaining electrospinning parameters [28].

The nanofibers morphology can be also adapted by changing the flow rate of electrospun 16% CA solution overstudied range (0.2–1 mL/h) with the same previously stated electrospinning parameters (20 kV applied voltage and 12 cm distance between needle tip to the collector). The SEM images of the different produced nanofibers shown in Figure 1(B1) observed that as the flow rate increased, the thickness of produce fibers was increased with the appearance of some droplets. This result may be attributed to the improvement in the solution flow rate, which delivers more polymeric composite solution to the nanofiber sheet, which increases the diameter of the liquid faucet and subsequently, the creation of thicker nanofibers [29]. Therefore, 0.2 mL/h was considered as the optimum flow rate for spinning 16% CA solution.

Another important factor in the fabrication of uniform electrospun nanofibers is the applied electrical voltage between the needle tip and collector. Three different electrical voltages (10, 18, 30 kV) were applied for electrospun 16% CA solution at the pre-optimized electrospinning conditions (0.2 mL/h flow rate and 12 cm distance between needle tip to the collector). The micrographs of the three different fabricated nanofibers at studied electrical voltages were illustrated in Figure 1(C1). It was indicated that the increment of applied voltage from 10 to 30 kV has a positive impact on decreasing the fiber diameter. This is due to the increase of the applied voltage, which enhances the magnitude of the electric field and results in a thinner liquid jet. Therefore, the optimum applied voltage was selected as 18 kV.

Three various distances of 8, 12 and 18 cm between the needle tip and collector were studied at a constant CA polymer concentration of 16%, at a spinning flow rate of 0.2 mL/h and 18 kV applied voltage [29]. The SEM images and the corresponding diameter dependence on distance are shown in Figure 1(D1). The distance from the needle tip to the collector has to be set to a golden distance to produce thinner fibers because any distance below or above this golden distance will result in larger fiber diameter. This is because, if the distance is too short, the fibers will not have enough time to dry and the fiber diameter increased. However, when the distance is too long, the effect of voltage on the liquid jet will be small, and hence results in much thicker fibers. In this investigation, it was observed that 18 cm is such a golden distance to produce good fibers and it was selected as the optimum distance.

Finally, the amount of ZnO NPs as filler to enhance the photocatalytic activity of nanofiber was optimized at the pre-optimized electrospinning conditions of nanofibers (0.2 mL/h solution flow rate, applied voltage 18 kV, polymer concentration 16% and distance between needle tip to the collector was set at 18 cm). The four different ZnO polymeric composite nanofibers at various filler amounts of 0.02, 0.05, 0.07, 0.1 wt% are imaged in Figure 1(E1). It was indicated from SEM images that the sample loaded with smaller ZnO filler (0.02%) showed homogeneous and uniform nanofibers without any beads but with a large fiber diameter range. As the amount of dispersed ZnO filler increased up to 0.05%, the diameter of produced fiber was decreased. This may be due to the properties of the semiconductors of immobilized ZnO that enhance the spinning properties of composite nanofiber. However, as the filler amount improved above 0.05%, the fiber diameter was increased and the beaded fiber was produced, owing to the formation of ZnO aggregates that hinder the spinning process and decrease the chance for the production of homogeneous nanofibers at higher ZnO filler concentrations [29,30]. Accordingly, 0.05% nano-ZnO was selected as the most proper filler concentration for the fabrication of a homogeneous and uniform ZnO polymeric composite nanofibers matrix with good porosity, as evident from the optical image of the matrix (Figure 1) with the average fiber diameter ranged between 125 and 170 nm.

### 3.2. Silver Immobilization onto ZnO Polymeric Composite Nanofibers for Enhancing Photocatalytic Activity

In order to improve the photocatalytic activity of fabricated ZnO polymeric composite nanofibers, it was surface-immobilized with Ag to decrease the fiber bandgap and subsequently improve its photocatalytic efficiency. This is mainly due to increased charge-transfer capacity by lowering bandgap energy; there was a minimized recombination of the excited electron-hole pairs of ZnO with the addition of Ag into the ZnO crystal lattice [31]. TEM images of ZnO polymeric nanofibers before (A) and after (B) silver immobilization are compared in Figure 2. In the case of ZnO polymeric nanofibers before Ag immobilization, a TEM image showed the homogeneous distribution of ZnO nanoparticles all over the fabricated nanofibers, while in the case of ZnO polymeric nanofibers after Ag immobilization, much higher intensity surface-immobilized nanoparticles were noticed that corresponded to Ag nanoparticles localized at the surface of the nanofibers. Moreover, dark agglomerated particles appeared for Ag-immobilized nanofiber; this may be due to the surface deposition of Ag nanoparticles onto ZnO nanoparticles loaded onto polymeric nanofiber that appeared as dark agglomerated spots.

### 3.3. Comparable Investigation of the Prepared ZnO Polymeric Composite Nanofibers before and after Silver Immobilization

#### 3.3.1. X-ray Diffraction Patterns (XRD)

In order to confirm the fabrication of ZnO polymeric composite nanofiber before and after silver immobilization, XRD patterns of various prepared materials including ZnO NPs, Ppy NPs and ZnO polymeric nanofibers after and before Ag immobilization were compared. Figure 3A showed the XRD pattern of ZnO NP synthesized using plant leaves extract; the pattern confirmed that the prepared material of wurtzite hexagonal structure and has a high crystalline degree. It indicated that distinctive peaks appeared at 31.7 (100), 34.4 (002), 36.2 (101), 47.5 (102), 56.6 (110), 62.8 (103), 67.9 (112) and 69.1 (201). This result is in perfect agreement with wurtzite ZnO (JCPDS ZnO Card No: 36_1451). The average particle size of ZnO NPs was calculated using the Scherrer equation that considers its nano-size range from 8.5 to 13 nm [32].

The XRD pattern of Ppy NPs synthesized via chemical oxidation polymerization illustrated in Figure 3B shows the broad peaks at the 2θ region 18–30°, revealing the amorphous nature of prepared Ppy NPs [33,34].

X-ray diffraction pattern of ZnO polymeric composite nanofibers investigated three sets of different diffraction peaks that corresponding to cellulose acetate, ZnO and Ppy. Figure 3C shows characteristic peaks for the wurtzite structure of ZnO at 2θ region of 31.7° (100), 34.4° (002), 36.2° (101), 47.5° (102), 56.6° (110), 62.8° (103), 67.9° (112) and 69.1° (201) in addition to cellulose acetate, which is well-defined at the crystalline peaks at 2θ equals to 13.83° and 16.63° which verified its crystalline nature. Finally, the broad peaks at 2θ of 18°–30°, revealing the presence of amorphous Ppy. This XRD pattern confirms the successful fabrication of ZnO polymeric composite nanofiber.

Regarding ZnO polymeric composite nanofiber after silver surface modification, the XRD pattern illustrated in Figure 3D showed most of the previously characteristic peaks of ZnO, Ppy and cellulose acetate with a reduction in peaks intensity. However, new peaks were illustrated at 37.8 (111), 43.8 (200) and 77.2 (311), corresponding to silver immobilized on the nanofiber surface [35,36]. The percentage of immobilized silver was detected using EDX analysis of ZnO polymeric composite nanofiber after silver immobilization; it was indicated that the weight % of the fiber analysis was oxygen 59.43, carbon 23.92, zinc 14.6 and silver 2.05. Accordingly, silver was immobilized, with around 2% by weight from the fabricated ZnO polymeric composite nanofibers.

#### 3.3.2. FTIR-Spectrophotometric Analysis

FTIR spectrums of the polymeric composite nanofiber before and after silver immobilization are illustrated in Figure 4A,B; both spectrums showed stretching peaks for N-H and C-H deformation vibration that were located at 1026 cm^−1^ and 1051 cm^−1^, respectively. The peaks at wavenumbers of 1051 cm^−1^, 1730 cm^−1^, 1735 cm^−1^ and 2947 cm^−1^ were attributed to O-CH_3_ stretching, C=O stretching, CH_3_ stretching and CH stretching vibrations, respectively.

A wide-range absorption band at 3483 cm^−1^ corresponding to N-H ring vibrations of polypyrrole at the composite nanofiber matrices was investigated. The characteristic bands in the range of 400–600 cm^–1^ are attributed to Zn–O stretching of ZnO material. These two FTIR spectrums confirm the fabrication of ZnO polymeric composite nanofibers. Comparing the two FTIR spectrums, it is clear that the silver surface-immobilized ZnO polymeric composite nanofiber characteristics by new peaks that appeared at 3483, 1739, 1535, 462 and 1404 cm^−1^. These new absorbance bands were attributed to the stretching vibrations of primary and secondary amines groups that confirm the presence of plant leaves extract used for the green synthesis of Ag on the nanofiber surface. Moreover, the characteristic peak, appearing in the range of 3300–3500 cm^‒1^, was broader at Ag immobilized nanofiber compared with the nanofiber before Ag immobilization; this may have originated from polyphenols stretching of the OH group that are present in the leaves extract after silver immobilization [20,37].

#### 3.3.3. Thermo-Gravimetric Analysis (TGA)

Figure 5A,B showed typical TGA analysis for ZnO polymeric composite nanofibers before and after silver immobilization.

For comparative purposes, the two TGA patterns showed almost the same multi-stage degradation steps. Both patterns showed slight mass losses of ~5% up to 250 °C, corresponding to the loss of volatile compounds and physically and chemically adsorbed water molecules bound to the hydrophilic (OH) groups of the cellulose acetate chains. These mass losses were followed by two more stages of thermal decomposition for the two patterns. The first step was between 250 and 410 °C, with mass losses ranged from ~70 to 75% for the two samples. This degradation step corresponds to the main thermal decomposition event and can be attributed to cellulose acetate chain degradation due to the breakdown of glycosidic bonds followed by the primary decomposition in volatile and dehydrated compounds. This degradation step was followed by the second degradation step between 410 and 500 °C, corresponding with nanofiber matrix carbonization, resulting in the complete degradation and decomposition of the polymeric matrix. This last step of thermal degradation highlighted the role of the presence of Ag surface immobilization in enhancing the thermal degradation performance of the ZnO polymeric composite nanofibers matrix [21].

Comparing the two patterns (Figure 5A,B) at this last thermal degradation step, it was indicated that the ZnO polymeric composite nanofibers matrix after Ag surface immobilization increases the thermal degradation to start at 416 °C compared with 406 °C for ZnO polymeric composite nanofibers matrix before Ag immobilization. Furthermore, the matrix weight loss was decreased to its half value for the Ag immobilized matrix. The weight losses were decreased from 4.6% for the ZnO polymeric composite nanofiber matrix before Ag immobilization to only 2.8% for that after Ag immobilization. These results confirm the positive role of Ag in improving the thermal stability of the ZnO polymeric composite nanofibers matrix.

### 3.4. Photocatalytic Characteristics of ZnO Polymeric Nanofibers for Malachite Green Dye Degradation

The performance of the two prepared ZnO polymeric nanofibers before and after Ag immobilization on malachite green (MG) dye photocatalytic degradation was compared against the different studied processing parameters.

#### 3.4.1. Effect of Contact Time

The effect of contact time for photocatalytic malachite green dye degradation as a function of contact time at the studied range between 0 and 120 min using 0.1 g from either ZnO polymeric nanofibers after or before Ag immobilization at room temperature is investigated in Figure 6.

It was demonstrated that MG degradation activity increased with increasing contact time and reached its maximum value after 60 min for both studied nanofiber matrices. Comparing the photocatalytic degradation performance of the two nanofiber matrices after and before Ag immobilization, it was evident that the degradation efficiency was upgraded from 63% before Ag immobilization to 93.5% at 60 min after Ag immobilization. The degradation profiles for both nanofiber matrices showed a slight increment at the dye degradation performance after 60 min up to 120 min, confirming that 60 min is suitable for completeness of the dye degradation process [12,38].

#### 3.4.2. Effect of Malachite Green Dye Concentration

The effect of malachite green dye concentration in the photocatalytic degradation process was studied by changing the initial dye concentration over the studied range of 10–120 ppm to assess the appropriate dye concentration at a constant nanofiber dose of 0.1 g and agitation speed of 300 rpm for 1 h at room temperature. It was indicated from Figure 7 that photocatalytic degradation efficiency decreased from 93.5 to 39% and 68 to 10% as the dye concentration increased from 10 to 120 ppm for both ZnO polymeric nanofibers after and before silver immobilization, respectively. This behavior may be due to the increment in the initial dye concentrations which increases the availability of more dye molecules for excitation and energy transfer. This dependence is perhaps related to the formation of several monolayers of adsorbed dye onto both ZnO polymeric nanofibers after and before silver immobilization, which is favored at high dye concentrations. Until the critical level was reached, the surface was not completely covered, leading to constant reaction rates [39]. After this critical level, with a further increment in the initial dye concentration, a significant decline in the degradation efficiency with the increase in dye concentration occurs due to several reasons. As the initial concentrations of the dye increase more and more, dye molecules are adsorbed on the surface of the photocatalytic materials and a significant amount of UV is absorbed by the dye molecules rather than the ZnO polymeric nanofibers matrix. Hence, the penetration of light to the surface of the photocatalytic decreases hydroxyl radicals generation, which declines the dye degradation efficiency. Moreover, the adsorbed dye on the photocatalyst matrix inhibits the reaction of adsorbed molecules with the photo-induced positive holes or hydroxyl radicals, since there is no direct contact of ZnO semiconductor or even Ag metal with them. Accordingly, as the initial concentration of dye increases, the requirement of catalyst surface needed for the degradation should be increased to offer sufficient hydroxyl radicals attacking the available dye molecules for the complete dye degradation process [40].

#### 3.4.3. Effect of Polymeric Nanofiber Dosage

The effect of polymeric nanofiber dosages on malachite green dye degradation over the studied range (0.05–0.25 g) after and before silver immobilization are compared in Figure 8. It was indicated that the two photocatalytic nanofibers were recorded at a maximum dye degradation efficiency when using 0.2 g. However, as the nanofiber dosages increased above 0.2 g, the dye degradation efficiency tend to decline. Therefore, the amount of photocatalytic studied nanofibers after and before Ag immobilization has both positive and negative impacts on the photodecomposition rate. In principle, the positive impact explained as the amount of catalyst is proportional to the number of active radicals like OH• and O_2_• which rapidly degraded the adsorbed MG dye [41]. However, this regime may be due to the number of photocatalytic nanofibers which increased up to 0.2 g, which increased the number of photons absorbed and consequently, the dye degradation rates increased. However, as the amount of nanofibers increases above 0.2 g, this increases the solution opacity and leads to a decrease in the penetration of the photon flux in the reaction media and thereby decreases the dye photocatalytic degradation rate. Comparing the photocatalytic profiles of the two ZnO polymeric nanofibers before and after Ag immobilization, a high increment in dye degradation efficiency was recorded for nanofiber before Ag immobilization compared with that after Ag immobilization, as the nanofiber dosage increased from 0.1 to 0.2 g. The dye degradation efficiency ranged from 55 to 96% compared with 93.5 to 97% as the nanofiber dosage improved from 0.01 to 0.2 g before and after Ag immobilization, respectively. These results confirm the dye degradation efficiency of ZnO polymeric nanofibers after Ag immobilization compared with the free one [42].

#### 3.4.4. Effect of Solutions pH

The solution pH modifies the electrical double layer of the solid electrolyte interface and consequently affects the sorption–desorption processes and the separation of the photogenerated electron–hole pairs in the surface of the semiconductor particles. Accordingly, the influence of different solution pH values ranged from 2 to 12 as the degradation efficiency of MG dye was investigated on ZnO polymeric nanofibers after and before silver immobilization, as shown in Figure 9. It was observed that the degradation efficiency of MG was increased with an increase in solution pH for the two studied ZnO polymeric nanofibers after and before silver immobilization. The probable reason for this behavior may be explained due to the increase of the electrostatic adsorption of cationic MG dye on the ZnO polymeric catalyst surface that dependent on its surface charge [43]. As the point of zero charges (z.p.c) of ZnO is equal to 6.4, so at pH < pH (z.p.c), the ZnO surface is positively charged (at acidic pH), which decreases the electrostatic attraction of cationic MG dye. Dissimilar at pH > pH (z.p.c), the ZnO surface is negatively charged (at pH > 6.4), which increases the electrostatic interaction between the negatively charged ZnO polymeric nanofibers and MG cations and leads to strong adsorption with a correspondingly high rate of dye degradation [44].

#### 3.4.5. Effect of Reaction Temperature

The effect of photocatalytic reaction temperature studied ranged from 25 to 80 °C on the degradation of malachite green dye using ZnO polymeric nanofibers after and before silver immobilization, as explained in Figure 10.

It was observed that the increase in the reaction temperature up to 50 °C resulted in an increase in the photodegradation efficiency of MG dye on the two ZnO polymeric nanofibers. This behavior may be returned to the increase in the reaction rate constant with increasing reaction temperature. On the contrary, as the reaction temperature increased above 50 °C and tended to the boiling point of water, the photodegradation efficiency of MG dye on the two ZnO polymeric nanofibers tend to decline [45]. This may be returned to the exothermic nature of the adsorption process of the reactant. At elevated temperatures, the adsorption of the reactant MG dye, which is a spontaneous exothermic phenomenon, is disfavored. Besides, increasing the temperature also disfavors the adsorption of the final reaction product, whose desorption tends to inhibit the reaction. Therefore, the optimum range of operational temperatures was found to be in the range of 35–50 °C [46].

### 3.5. Reusability of Fabricated ZnO Polymeric Composite Nanofibers

Economically, the ability of the photocatalytic materials to be reused represents one of the paramount characteristics of efficient catalytic materials. Therefore, the reusability of both of the spent ZnO polymeric composite nanofibers before and after silver immobilization after the dye decolorization process was investigated using HCl, and their ability for further reuse as photocatalytic membranes are represented in Figure 11. In order to confirm that the regenerated active sites at the regenerated ZnO polymeric composite membranes after and before silver immobilization have the ability for degrading MG dye molecules, the HCl-regenerated nanofibers membranes were tested for 10 cycles as photocatalytic membranes. It was indicated from Figure 11 that the dye decolorization efficiency showed a minor decline for the regenerated Ag/ZnO polymeric nanofibers from 97% (cycle 1) to 72% (cycle 10); they lost around 25% from their initial efficiency. However, the regenerated ZnO polymeric nanofibers lost around 28% of their initial photocatalytic efficiency. This result may be due to incomplete desorption of MG dye from the surface of nanofibers when repeatedly used after regenerated with HCl. Thus, this result confirms the efficient reusability and stability of prepared ZnO polymeric composite as photocatalytic membranes after its regeneration with HCl up to 10 cycles. Consequently, the fabricated ZnO polymeric nanofibers after and before Ag immobilization may be classified as economical and efficient photocatalytic membranes, which are crucial for industrial and large-scale practical applications.

### 3.6. Suggested Mechanism of Malachite Green Dye Degradation Using ZnO Polymeric Nanofibers

In general, photocatalytic degradation involves several steps such as adsorption–desorption, electron–hole pair production and the recombination of the electron pair and chemical reaction. The suggested mechanism for photocatalytic degradation of organic molecules such as MG dye is explained in Figure 12.

When ZnO polymeric nanofiber is irradiated with continuous photons energy similar to or more than the ZnO bandgap, the electrons (e^−^) are more excited and transferred from the ZnO polymeric catalyst valence band (VB) into the ZnO polymeric catalyst conduction band (CB) with the spontaneous creation of holes (h^+^) in the valance band as follows:ZnO polymeric nanofiber photocatalyst + hv → e CB^−^ + h VB^+^(2)

The substantial energy required to transfer the electrons from the VB into the CB is expressed with hV. The electrons generated during the radiation can be easily trapped by O2 that is absorbed on the solved superoxide or the ZnO polymeric nanofiber photocatalyst surface to give radicals of superoxide (O_2_^−^).
e CB^−^ + O_2_ → O_2_^−^(3)

Therefore, O_2_^−^ could react with H2O molecules to produce hydroperoxy radical (HO_2_) and hydroxyl radical (OH), which are strong oxidizing agents, to decompose the organic molecule of MG dye [47,48,49,50].
O_2_^−^ + H_2_O → HO_2_∙+ OH(4)

The photogenerated holes must be trapped by surface (OH or H_2_O) on the photocatalyst surface to give hydroxyl radicals (OH):h VB^+^ + OH^−^ → OH∙_ad_(5)
h VB^+^ + H_2_O → OH + H^+^(6)

The organic molecules of MG dyes will be oxidized and converted to CO_2_ and H_2_O according to the following equation:OH∙+ organic molecules (MG dye) + O_2_ → products (CO_2_ and H_2_O)(7)

Meanwhile, the recombination of the positive hole and electron could take place, which could reduce the photocatalytic activity of the prepared ZnO polymeric nanofiber. This limitation was solved through surface modification of ZnO polymeric nanofiber with nanosilver to act as electron sinks to avoid the recombination process.
e CB^−^ + h VB^+^ → PC(8)

ZnO cannot absorb much of the sunlight directly due to its large bandgap of 3.37 eV, but colored organic MG dye adsorbed on its surface and acts as a photosensitive matrix. Besides, the inclusion of the polymeric matrix with semiconductor polypyrrole (PPy) polymer with the π-orbital, which possesses highest occupied molecular orbital (HOMO) in the combined system, found that the lowest unoccupied molecular orbital (LUMO) levels of the polymer were energetically higher than the conduction band edge of ZnO [47]. Therefore, the suggested electron transfer paths for MG dye degradation are illustrated in Figure 11. As a result, rapid-charge separation and slow-charge recombination may occur using silver immobilized ZnO nanofiber.

When the ZnO polymeric composite nanofibers were illuminated under visible light, their electrons can be excited from the HOMO to the LUMO at the polymeric matrix, whereas the holes were left at the HOMO of the polymer. The excited-state electrons can be readily injected into the CB of ZnO, and then further injected into the Ag fermi level. The metallic Ag nanoparticles functioned as an electron sink to accept the photo-induced electrons from the excited semiconductor. Silver-immobilized ZnO nanofiber had a quicker charge separation and slower charge recombination process compared with ZnO polymer nanofiber before Ag immobilization and thus possess higher photocatalytic activity. Moreover, the presence of the PPy polymer may cover the surface of ZnO and form a relatively thick layer that hinders the injection of excited electrons from the outer PPy polymer layer to the inner ZnO layer. Consequently, OH· radicals decreased and the photodecomposition of the target contamination was affected [51].

## 4. Conclusions

In this work, 0.05% nano-zinc oxide was impregnated as a filler onto composite polymeric membrane composed from semiconductor polypyrrole mixed with cellulose acetate to fabricate homogeneous and uniform nanofiber using electrospinning at a 0.2 mL/h solution flow rate, applied voltage of 18 kV, polymer concentration of 16% and the distance between needle tip to the collector were set at 18 cm. As an attempt to increase the photocatalytic efficiency, the surface of this fabricated nanofiber was modified through immobilization with a nanosilver. The photocatalytic efficiency of the two fabricated nanofibers before and after silver immobilization was compared for malachite green dye degradation. It was evident that the silver-immobilized nanofiber showed higher photocatalytic degradation efficiency compared with that before silver immobilization for all studied degradation processing parameters including dye concentration, photocatalysis dosage, solution pH and reaction temperature. Immobilized nanosilver acted as an electron sink to avoid electron recombination that enhances photocatalytic activity. It was indicated that the increment at both photocatalysis dosage and solution pH has a positive impact on the MG dye degradation efficiency. On the contrary, the increases in dye concentration and reaction temperature have a negative impact on MG dye degradation efficiency.

## Figures and Tables

**Figure 1 polymers-13-02033-f001:**
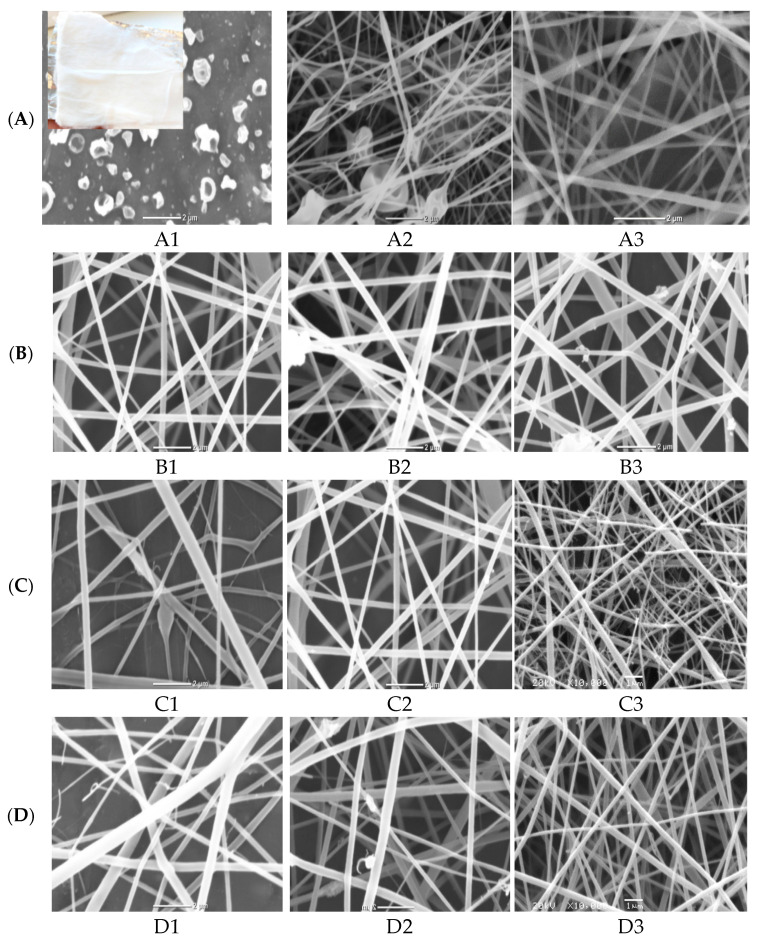
SEM micrographs of electrospinning processing parameters on fabricated ZnO polymeric composite nanofiber: (**A**) Cellulose acetate concentration (8–16 wt%), (**B**) solution flow rate (0.2–1 mL/h), (**C**) applied voltage (10–30 KV), (**D**) distance between needle tip and collector (8–18 cm) and (**E**) amount of loaded ZnO NPs as a filler (0.02–0.1 wt%).

**Figure 2 polymers-13-02033-f002:**
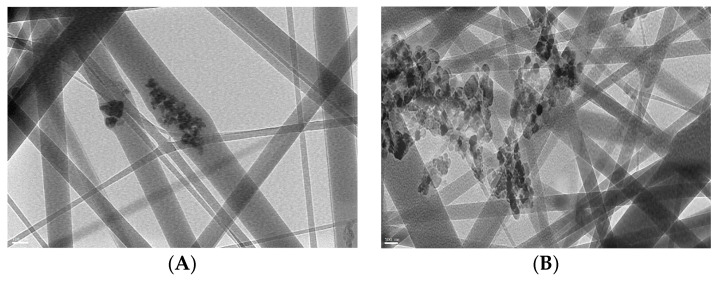
TEM micrographs of ZnO polymeric composite nanofiber (**A**) before silver immobilization and (**B**) after silver immobilization.

**Figure 3 polymers-13-02033-f003:**
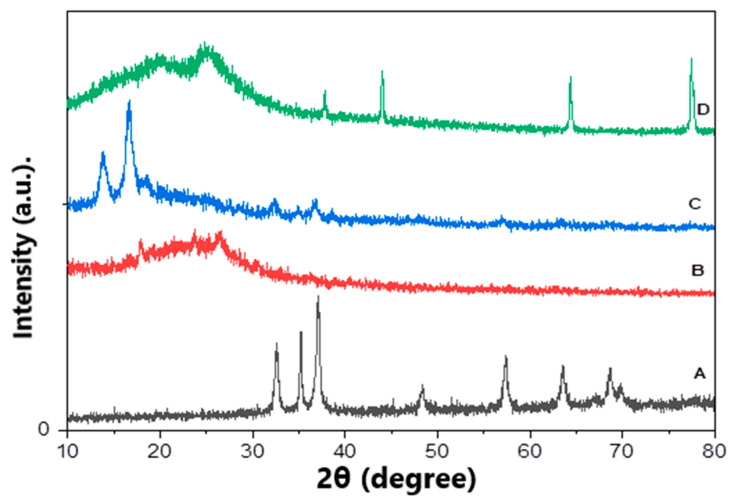
XRD patterns of various prepared matrices, (**A**) ZnO NPs, (**B**) Ppy NPs, (**C**) ZnO polymeric composite nanofibers and (**D**) ZnO polymeric composite nanofibers after silver immobilization.

**Figure 4 polymers-13-02033-f004:**
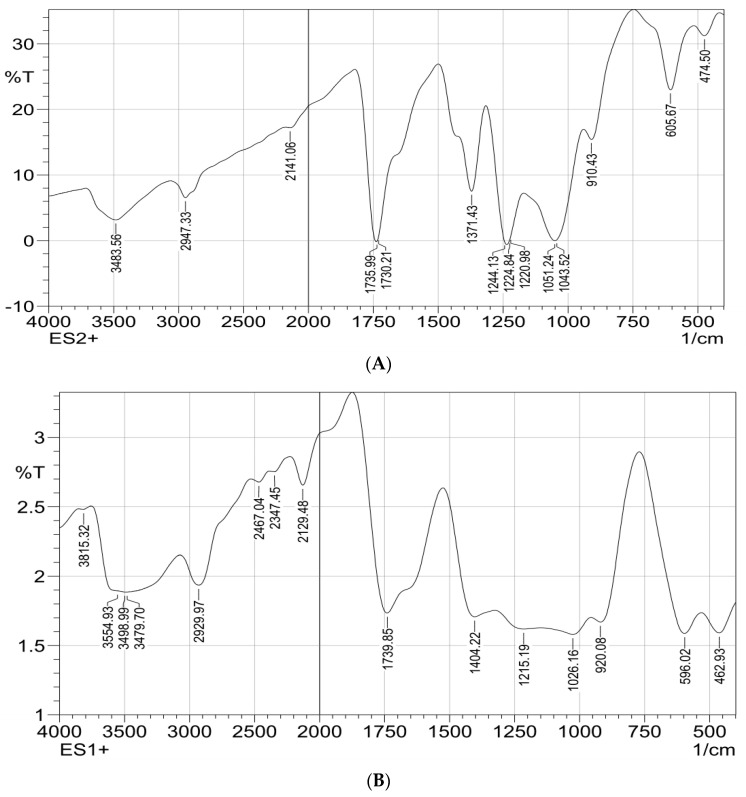
FTIR spectrophotometric analysis for ZnO polymeric composite nanofiber (**A**) before silver immobilization and (**B**) after silver immobilization.

**Figure 5 polymers-13-02033-f005:**
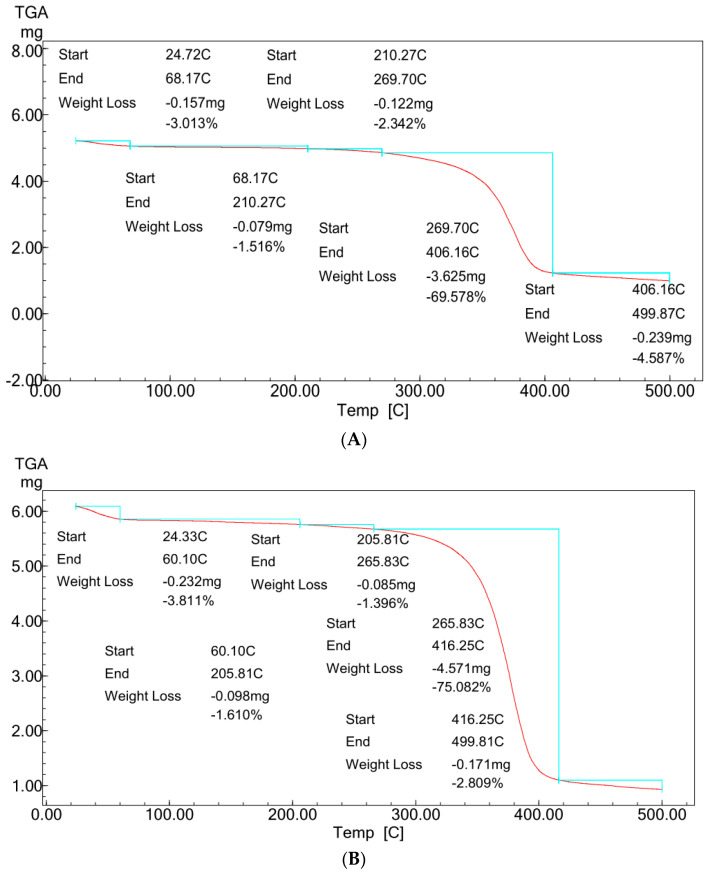
TGA profiles of ZnO polymeric nanofiber (**A**) before silver immobilization and (**B**) after silver immobilization.

**Figure 6 polymers-13-02033-f006:**
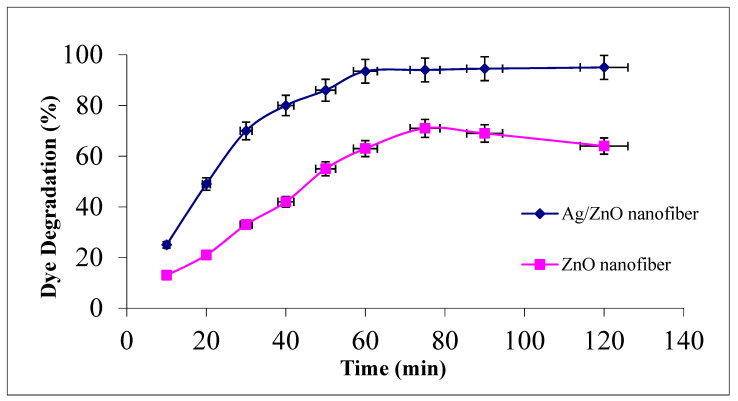
Effect of contact time on MG dye photocatalytic degradation process using ZnO polymeric nanofiber after and before silver immobilization (initial dye concentration = 20 ppm, nanofiber dose = 0.1 g, pH = 7 at room temperature).

**Figure 7 polymers-13-02033-f007:**
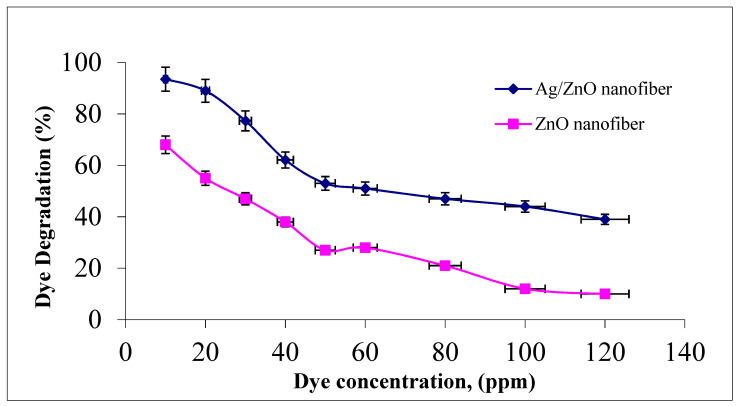
Effect of initial MG dye concentration on MG dye photocatalytic degradation process using ZnO polymeric nanofiber after and before silver immobilization (contact time = 60 min, nanofiber dose = 0.1 g, pH = 7 at room temperature).

**Figure 8 polymers-13-02033-f008:**
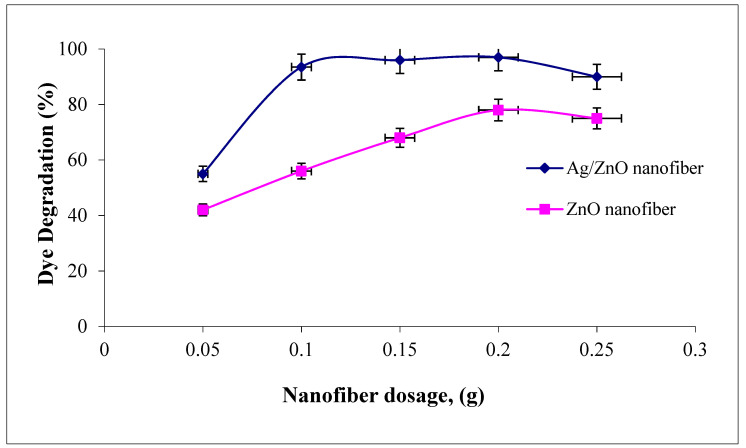
Effect of adsorbent dosage on the MG dye photocatalytic degradation process using ZnO polymeric nanofibers after and before silver immobilization (initial dye concentration = 20 ppm, pH = 7, contact time = 60 min at room temperature).

**Figure 9 polymers-13-02033-f009:**
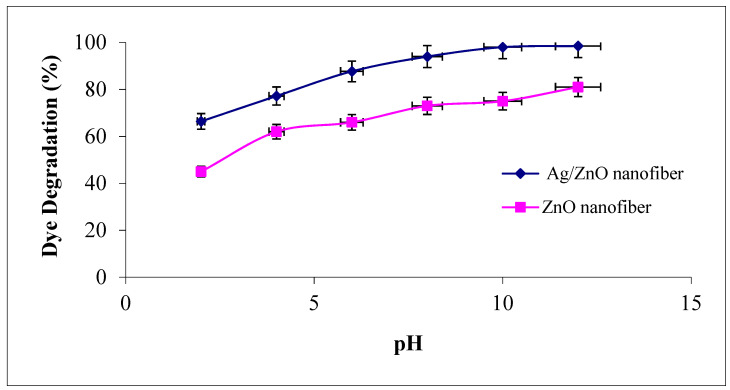
Effect of dye solution pH on the MG dye photocatalytic degradation process using ZnO polymeric nanofibers after and before silver immobilization (initial dye concentration = 20 ppm, nanofiberdose = 0.1 g, contact time = 60 min at room temperature).

**Figure 10 polymers-13-02033-f010:**
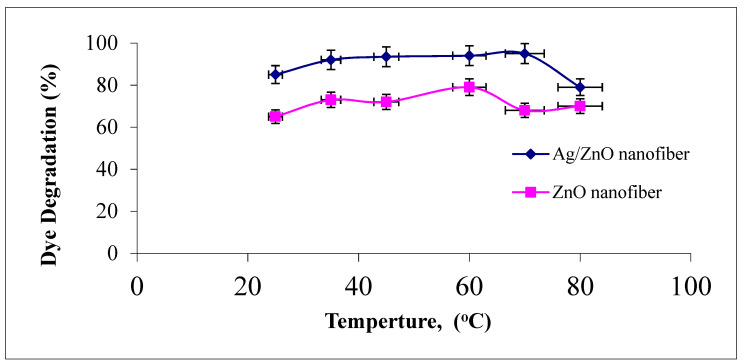
Effect of dye solution temperature on the MG dye photocatalytic degradation process using ZnO polymeric nanofibers after and before silver immobilization (initial dye concentration = 20 ppm, nanofiberdose = 0.1 g, contact time = 60 min at pH = 7).

**Figure 11 polymers-13-02033-f011:**
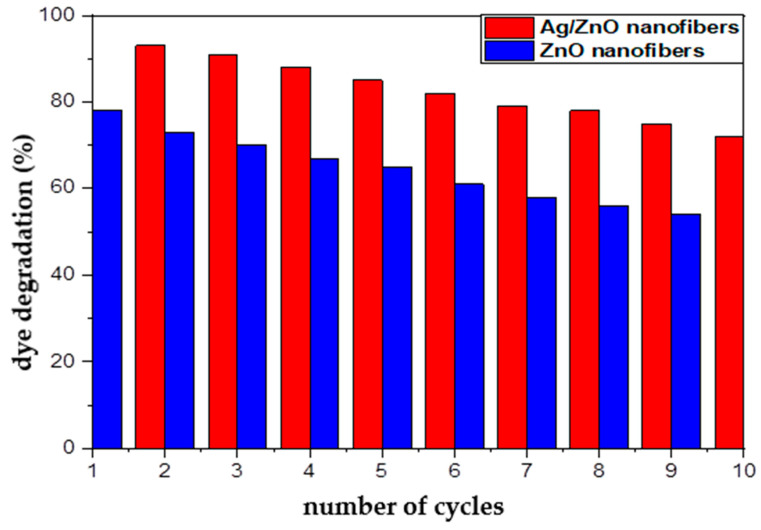
Reusability of ZnO polymeric composite nanofibers before and after silver immobilization as photocatalytic materials for MG dye decolorization process.

**Figure 12 polymers-13-02033-f012:**
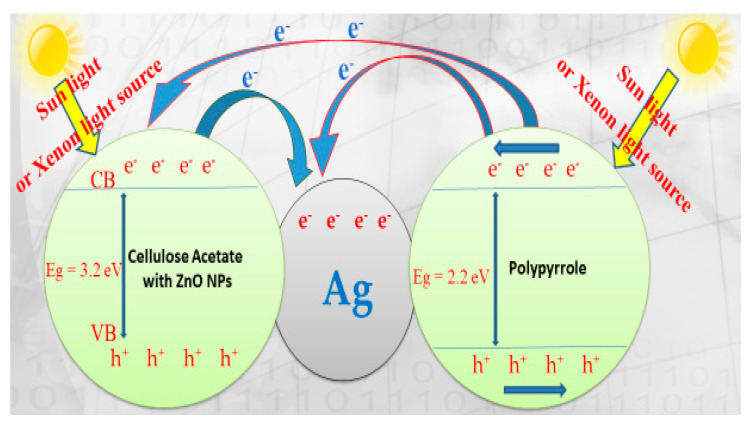
OH^+^ Malachite green dye → products (CO_2_ and H_2_O). Scheme diagram of the photocatalytic degradation mechanism of malachite green dye (MG) on ZnO polymeric nanofiber immobilized with silver.

## Data Availability

Not applicable.

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
