# Peer review of "Photocatalytic Degradation of Malachite Green Dye from Aqueous Solution Using Environmentally Compatible Ag/ZnO Polymeric Nanofibers"

_polymers, 2021, doi:10.3390/polym13132033_

Round 1

Reviewer 1 Report

In this work, the author designed and synthesized a ZnO/CA/Ppy polymeric composite nanofiber for photocatalytic degradation of dye. The micro-morphology and physicochemical properties of the catalyst were characterized. And the performance of this hybrid was evaluated. Topic of this study is of interest to the reader of Polymer. Therefore, I recommend the acceptance of this work after some revisions. Some specific comments are shown below.

  1. Error bars are lacking in the figures. Some important degradation experiments should be performed in duplicate or triplicate and the results were presented with the mean data with standard deviations.
  2. Figure 3: the resolution of this image is too low to be accepted for publication
  3. The reactive oxygen species (ROSs) generated in reaction system should be identified by the trapping experiments to explore the possible reaction mechanism. The experimental details can refer and cite these manuscripts: Environ. Sci. Technol., 2020, 54, 8464-8472; Adv. Funct. Mater., 2018, 28, 1705295; J. Hazard. Mater., 2017, 322: 532-539.
  4. It is better to provide the mineralization efficiency of dyes. Besides, colorless organic contaminants, such as phenol, could be a better target pollutant in the evaluation of catalyst performance.
  5. The stability and durability of as-synthesized catalyst should be evaluated by the cycling tests.

Author Response

Reviewer #1:

Q1- Error bars are lacking in the figures. Some important degradation experiments should be performed in duplicate or triplicate and the results were presented with the mean data with standard deviations.

All the dye degradation experiments were repeated triplicate and the mean data were investigated at the figures with the standard deviations.  

Q2- Figure 3: the resolution of this image is too low to be accepted for publication.

Figure 3 was redraw again with high resolution. 

Q3- The reactive oxygen species (ROSs) generated in reaction system should be identified by the trapping experiments to explore the possible reaction mechanism. The experimental details can refer and cite these manuscripts: Environ. Sci. Technol., 2020, 54, 8464-8472; Adv. Funct. Mater., 2018, 28, 1705295; J. Hazard. Mater., 2017, 322: 532-539.

Thank you for your comment, the equilibrium, kinetics and complete discussed photocatalytic mechanism of this prepared novel ZnO/CA/Ppy hybrid nanofibers matrix were discussed at the second paper that complete this work and will be submitted recently for publication. We targeted to divide this work into 2 papers, the first one concerned with investigation the novel material preparation and characterization and testing its photocatalytic behaviour against the various processing parameters with brief suggestion of photocatalytic mechanism. The second will concerning with the equilibrium, kinetics and thermodynamics of the photocatalytic reaction including detailed discussion about the photocatalytic mechanism through experimental detection and generation of reactive oxygen species during photocatalysis process.

The suggested references were added at the section deals with brief mechanism and these references also were added at the prepared second paper that deals with complete discussion of the photocatalytic mechanism.    

Q4-  It is better to provide the mineralization efficiency of dyes. Besides, colorless organic contaminants, such as phenol, could be a better target pollutant in the evaluation of catalyst performance.

We select malachite green (MG) dye as the targeted pollutant to test the photocatalytic degradation efficiency of the novel prepared ZnO/CA/Ppy hybrid nanofibers matrix through dye decolorization using spectrophotometric analysis. However, the final products from this degradation was investigated in details at the second paper that deals with the photocatalytic degradation mechanism through oxygen generation that was proved using HPLC analysis to confirm that the MG dye was firstly degraded into 4-diaminophenol and benzaldehyde through C–C bond cleavage then 4-diaminophenol further degraded into hydroquinone by the action of deamination. This hydroquinone could be further degraded into CO2 and water as final degradation products. All these results were included at the second paper that represents completeness of this work.  

Q5- The stability and durability of as-synthesized catalyst should be evaluated by the cycling tests.

The stability and durability of fabricated ZnO polymeric composite nanofibers before and after silver immobilization was studied experimentally and the data was added a the modified manuscript version.

Reviewer 2 Report

I have read with interest the manuscript from Elkady et al. I personally believe the authors have produced a nice work, demonstrating an example of printable photocatalytic ink produced by a biocompatible process through optimization of the fabrication process. The manuscript could be accepted with major revisions given these modifications:

  1. The authors should provide some bibliographic references demonstrating the beneficial combination of Ag and ZnO nanomaterials
  2. “Decolonization efficiency” could be corrected to “Decoloration efficiency”.
  3. How do the authors estimate porosity of their composite material? It is not clear how the samples look, an optical investigation would be very useful.
  4. Can the authors report an EDX characterization of the samples to demonstrate the presence of silver and zinc in their material?
  5. Some figures of the manuscript are of low quality. A particular concern with XRD characterization (Figure 3). The authors should report good quality figures for the XRD patterns, along with clear indications of the peaks. Also Figure 11 should be significantly improved.
  6. Figure 6-10 report on the effect of different parameters towards the composite photocatalytic activity. The authors should indicate the errors associated the measurements to assess the beneficial effect of Ag towards ZnO based photocalysis.

Author Response

Q1- The authors should provide some bibliographic references demonstrating the beneficial combination of Ag and ZnO nanomaterials.

The bibliographic reference of “Viet, T; The, C; Tri, P; Tien, P; Manh, L. Synergistic Adsorption and Photocatalytic Activity under Visible Irradiation Using Ag-ZnO/GO Nanoparticles Derived at Low Temperature. Journal of Chemistry. 2019, 2019, 1-13” was added that demonstrating the beneficial combination of Ag and ZnO nanomaterials and the discussion of section “Silver immobilization onto ZnO polymeric composite nanofibers for enhancing photocatalytic activity” was modified based on this reference.

Q2- Decolonization efficiency” could be corrected to “Decoloration efficiency”.

The word Decolonization was corrected.

Q3- How do the authors estimate porosity of their composite material? It is not clear how the samples look, an optical investigation would be very useful.

Thank you for comment, the porosity of the composite nanofiber matrix was expected from the optical view of the nanofiber matrix that was inserted at Figure 1 at the modified manuscript. 

Q4- Can the authors report an EDX characterization of the samples to demonstrate the presence of silver and zinc in their material?

The EDX analysis of ZnO polymeric composite nanofiber after silver surface modification was take place to elucidate the percentage of silver immobilization and added at the modified manuscript.

Q5- Some figures of the manuscript are of low quality. A particular concern with XRD characterization (Figure 3). The authors should report good quality figures for the XRD patterns, along with clear indications of the peaks. Also Figure 11 should be significantly improved.

Figures (3 &12) were modified and draw again to improve their resolutions.

Q6-The Figure 6-10 report on the effect of different parameters towards the composite photocatalytic activity. The authors should indicate the errors associated the measurements to assess the beneficial effect of Ag towards ZnO based photocalysis.

All the dye degradation experiments were repeated triplicate and the mean data were investigated at the figures (6-10) with errors associated the measurements.

Finally authors want to thank reviewers for their valuable comments which are improved the scientific value of the manuscript

Round 2

Reviewer 1 Report

The authors were revised the manuscript based on the reviewers remarks to improve the manuscript. This can be accepted.

Reviewer 2 Report

The authors have sufficiently solved all the raised questions. The paper is now acceptable for publication.